# An analysis of retracted papers in Computer Science

**Martin Shepperd**[1,2]* *, Leila Yousefi[3]*

**1** Department of Computer Science, Gothenburg|Chalmers University, Gothenburg, Sweden, **2** Department of Computer Science, Brunel University London, Uxbridge, United Kingdom, **3** Department of Life Sciences, Brunel University London, Uxbridge, United Kingdom

☯ These authors contributed equally to this work.
* martin.shepperd@gu.se, martin.shepperd@brunel.ac.uk

## Abstract

### Context

The retraction of research papers, for whatever reason, is a growing phenomenon. However, although retracted paper information is publicly available via publishers, it is somewhat distributed and inconsistent.

### Objective

The aim is to assess: (i) the extent and nature of retracted research in Computer Science (CS) (ii) the post-retraction citation behaviour of retracted works and (iii) the potential impact upon systematic reviews and mapping studies.

### Method

We analyse the Retraction Watch database and take citation information from the Web of Science and Google scholar.

### Results

We find that of the 33,955 entries in the Retraction watch database (16 May 2022), 2,816 are classified as CS, i.e., $\approx$ 8%. For CS, 56% of retracted papers provide little or no information as to the reasons. This contrasts with 26% for other disciplines. There is also some disparity between different publishers, a tendency for multiple versions of a retracted paper to be available beyond the Version of Record (VoR), and for new citations long after a paper is officially retracted (median = 3; maximum = 18). Systematic reviews are also impacted with $\approx$ 30% of the retracted papers having one or more citations from a review.

### Conclusions

Unfortunately, retraction seems to be a sufficiently common outcome for a scientific paper that we as a research community need to take it more seriously, e.g., standardising procedures and taxonomies across publishers and the provision of appropriate research tools.

**Data Availability Statement:** The complete raw data cannot be shared publicly because of a data usage agreement with Retraction Watch that prohibits publishing more than 2% of the data set. This requirement arises because, in order to fund Retraction Watch's continued operations, given that their initial grants have ended, they are

licensing their data to commercial entities. Therefore researchers will need to approach Retraction Watch directly (retractionwatch.org) to obtain the same data set. We have placed our relevant code for this study in the Zenodo repository (https://doi.org/10.5281/zenodo.6634462).

**Funding:** The author(s) received no specific funding for this work.

**Competing interests:** The authors have declared that no competing interests exist.

Finally, we recommend particular caution when undertaking secondary analyses and meta-analyses which are at risk of becoming contaminated by these problem primary studies.

## Introduction

With an ever-increasing number of scientific papers being published every year. For example, Elsevier alone published in excess of 550,000 new scientific articles in 2020 across roughly 4,300 journals; its archives contain over 17 million documents (2020 RELX Company Report). Similarly, the rising count of retracted papers has been noted (e.g., [1–3]). What is unclear is whether the growth is due to rising misconduct or improvements in our abilities to detect such situations as argued by Fanelli [2]. In addition, new questions arise concerning our ability to police or manage those papers that are retracted. This is a particularly important matter if such papers are subsequently cited to support some argument, or worse, contribute to, or contaminate, a meta-analysis so that weight is given to one or more primary studies that are no longer considered valid.

This problem is exacerbated by the increasing use of open archives, e.g., arXiv to publish pre- or post-prints of papers in addition to publishers' official publication websites—which are frequently protected by paywalls. The challenge here is that some archives are relatively informally organised and often rely on voluntary activities or temporary income streams. This means there may not necessarily be any formal mechanism for dealing with expressions of concern or retraction and even if such decisions are made, they might not propagate across all versions of the paper in question. Indeed, as we will show, they do not.

Although various other scientific disciplines have raised retraction as a concern this has not been explicitly investigated within the discipline of Computer Science (CS). The sole exception, to the best of our knowledge, is Al-Hidabi and Teh [4] who examined 36 CS retractions, constituting approximately 2% of those actually retracted at the time of their analysis (2017). Fortunately, we have been able to benefit from the pioneering work of Retraction Watch (RW) and their database [5] to enable a considerably more comprehensive plus up-to-date analysis of retraction in CS.

Therefore, we investigate the nature and prevalence of retracted papers in CS. In addition, we consider how such papers are cited, in particular how they are used by secondary analyses, systematic reviews and meta-analyses and their post-retraction citation behaviour.

The remainder of the paper is structured as follows. The next section provides general information about the retraction, reasons for retraction, a brief history and some indicators of the overall scale. We then describe our methods based on the Retraction Watch [5] and the Google Scholar citation databases and present the detailed results from our data analysis organised by the research question. We conclude, with a discussion of some threats to validity, a summary of our findings and the practical implications that arise from them.

## Background

We start with some definitions.

**retraction** means that a research paper has been formally removed from the scientific body of literature. This might occur for multiple reasons, but at a high level, it means that a research paper has been formally indicated as not for credible use as a reference in the scholarly literature. Note, however, that not all retraction reasons imply that the scientific results are

considered unreliable, e.g., retraction due to plagiarism. This can be for reasons ranging from honest error to scientific malpractice. Typically, but unfortunately not always, the article as first published is retained online in order to maintain the scientific record but some watermark or other modifier is applied so that the retraction and reasons for it are now linked to the paper.

**expression of concern** (EoC) may be issued by publishers or editors when there exist well-founded concerns regarding a research paper and it is believed that readers need to be made aware of potential problems that may require changes to the manuscript. An EoC can be updated to a retraction if the grounds for concern become more compelling.

**research misconduct** using the US Office for Research Integrity definition [6] suggests "fabrication, falsification, or plagiarism in proposing, performing, or reviewing research, or in reporting research results". Of course, a retraction does not necessarily imply misconduct.

**questionable research practice** (QRP) also called p-hacking in the context of the null hypothesis significance testing paradigm, includes analysing data in multiple ways but selectively reporting only those that lead to 'significant' results [7]. As such, this would not usually be seen as misconduct although it is generally poor scientific practice. Thus QRPs cover something of a spectrum ranging from what are sometimes referred to as 'honest errors' to more cynical and egregious errors [8].

**version of record** (VoR) is the final published version of a paper and so includes any editorial improvements e.g., the application of house style, running headers and footers including pagination that are made once the peer review process is complete. This will be distinct from a post-print which is the version of the paper after peer review and acceptance but before final copy editing on the part of the publisher, in other words, the final version handled by the author(s). Note that for conferences where the authors are expected to produce camera-ready versions of their papers, the difference between the post-print and the VoR can be negligible.

A research paper may be retracted for one or more reasons. Note that the list of reasons is growing over time as publishers and editors encounter new situations and find the need for new categories. RetractionWatch presently identify 102 reasons [9]. This is of course a very fine-grained classification scheme. A non-exhaustive list of the more commonplace reasons includes:

- plagiarism including self-plagiarism

- incorrect or erroneous analysis

- doubtful or fraudulent data

- mislabelled or faked images

- inappropriate or fraudulent use of citations

- lack of ethics approval obtained prior to commencement of the study

- fake peer review

- objections from one or more of the authors

- problematic or fake author(s)

Note that not all of the above reasons are related to research misconduct. So we should be clear that whilst retraction means something has gone wrong, it does not necessarily imply

culpability or some kind of moral lapse on the part of one or more of the authors. We should perhaps find encouragement in the findings of Fanelli [10] that only 1-2% of researchers admitted to ever fabricating data or results. In contrast, the survey of 2,000 experimental psychologists by John et al. [8] found that the use of so-called QRPs was disturbingly high. Thus, whilst misconduct is likely very rare, poor research practices may be less so.

A major contributor to our understanding of the phenomenon of retracted papers is the not-for-profit organisation RW founded by Adam Marcus and Ivan Oransky in 2010 who maintain the largest and most definitive database of retracted papers. This was kindly made available for analysis in this paper. For an overview of the role of the RW organisation see Brainard [11].

Investigating retracted papers is not new. More than 30 years ago Pfeifer and Snodgrass [12] looked into this phenomenon within medical research. They found 82 retracted papers and then assessed their subsequent impact via post-retraction citations which totalled 733. They computed that this constituted an approximate 35% reduction in what might otherwise have been expected but is still disturbing particularly in a critical domain such as medicine.

Within retraction, one topic that has been quite widely investigated is identifying possible predictors for papers likely to be retracted. However, the obvious predictor of the prevalence of negative citations (i.e., those questioning the target paper) was found to have surprisingly little impact on the likelihood of retraction by Bordignon [13]. More positively, Lesk et al. [14] reported that PLoS papers that included open data were substantially less likely to be retracted than others.

Another area of investigation has been the citation patterns of papers *after* they have been retracted. As mentioned, back in 1990 Pfeifer and Snodgrass [12] found that citations continued although at a reduced rate. More recently, Bar-Ilan and Halevi [15] analysed 15 retracted papers from 2015-16 for which there were a total of (obtainable) 238 citing documents. Given that the papers were all publicly retracted this is a disturbingly high level. However, the authors pointed out that not all reasons for retraction of necessity invalidate the findings e.g., self-plagiarism means the authors are unjustifiably attempting to obtain extra credit for their research, not that the research is flawed. Of the 15 papers, 8 constituted more serious cases where the data or the images were manipulated hence the results cannot be trusted. Of the citations to these papers that contained unsafe results 83% were judged to be positive, 12% neutral and only 5% were negative. This is despite being after the papers were publicly retracted, leading to concerns about the way this information is disseminated or the state of scientists' scholarship.

This analysis of post-retraction citation patterns was followed up by Mott et al. [16] and Schneider et al. [17] both examining the field of clinical trials and more recently, Heibi and Peroni [18] and Bolland et al. [19] looking more generally. Even secondary studies and systematic reviews are not immune e.g., the "Bibliometric study of Electronic Commerce Research in Information Systems & MIS Journals" (Note, we are intentionally not citing retracted papers but rather referring to them by their title. This enables the curious to locate them should they wish.) This review was published in 2016, retracted in 2020, and has still been cited 11 times subsequently. If systematic reviews are seen as the 'pinnacle' of the evidence hierarchy then we should be particularly vigilant about their possible contamination through the inclusion of retracted primary studies. Brown et al. [20] explored these phenomena in pharmacology and found that of 1,396 retracted publications in the field approximately 20% (283) were cited by systematic reviews. Of these, 40% were retracted for data problems including falsification, and 26% for concerns about methods and analysis.

Other relevant work has also been undertaken by researchers trying to understand the reasons for citations in scientific papers. Key themes are that citations need to be understood in context, that they cannot be seen as homogeneous, and that they are not "simply a function of

scientific argumentation in a pure sense, there are many motives for citing authors, a point hidden by, or only implicit in, many functionalist accounts of citation" Erikson and Erlandson [21]. This certainly seems to be borne out by the number of papers retracted for inappropriate citations.

Erikson and Erlandson [21] go on to suggest that overall motivation falls into one of four categories:

**argumentation** which might be seen as the traditional form of citation in the scientific literature where the citation is used to support a particular viewpoint.

**social alignment** where the motive for citing is driven by the author's identity or self-concept e.g., to show his or her erudition or perhaps their alignment with other groups of researchers.

**mercantile alignment** For instance, "giving credit to other people's work differs from including it in an argument" [21] whilst other less altruistic versions might include self-promotion and bartering when there is the hope that the cited author will cite him or her in return or the expectation that if the cited person happens to be a referee they will behave more generously.

**data** which is again closer to what one might expect in the scientific community where the citation is the source of the data, for instance, a data set or perhaps a systematic review or meta-analysis.

The complexities of actual citation practice, we believe, demonstrate the need for care when analysing real citation behaviour from published scientific papers. Interestingly, a number of researchers have sought to automate the process of determining citation purpose using a range of contextual data along with linguistic cues, e.g., Teufel et al. [22], more recently Heibi and Peroni [23], and in the specific domain of algorithm citation, Tuarob et al. [24]. This supports the notion that treating all citations as equal in quantitative citation analysis may be misleading. So it is likely that not all citations to retracted articles are going to be equally impacted, but those from the argumentation and data categories will tend to be most vulnerable.

Despite the foregoing discussion, there are almost no similar studies in Computer Science and none that examine the potential impact of systematic reviews, mapping studies, and meta-analyses. The only work we have located is by Al-Hidabi and Teh [4] which retrieved 36 retracted CS papers and classified the reasons as random (4/36), non-random (31/36), and no reason given (1/36). By random it would seem the authors are meaning so-called 'honest' errors and non-random refers to various forms of misconduct such as plagiarism and duplicate publication. However, we do feel constrained to observe that 36 out of 1818 retracted papers in the RW database at the point of their analysis (2017) is a quite small (2%) proportion.

## Analysis and results

In order to explore the phenomena of retracted scientific papers, we made use of the Retraction Watch database [5], dated 16th May 2022. This comprises 33,955 records, one per retracted paper, of which 2,816 are classified as Computer Science, i.e., approximately 8.6%. The RW data set covers papers in any scientific discipline (very broadly defined) and includes Business and Technology, Life Sciences, Environmental Sciences, Health Sciences, Humanities, Physical Sciences, and Social Sciences. In addition to bibliographic data for each retracted paper, the database contains the publication date, retraction date, and retraction reason(s) and

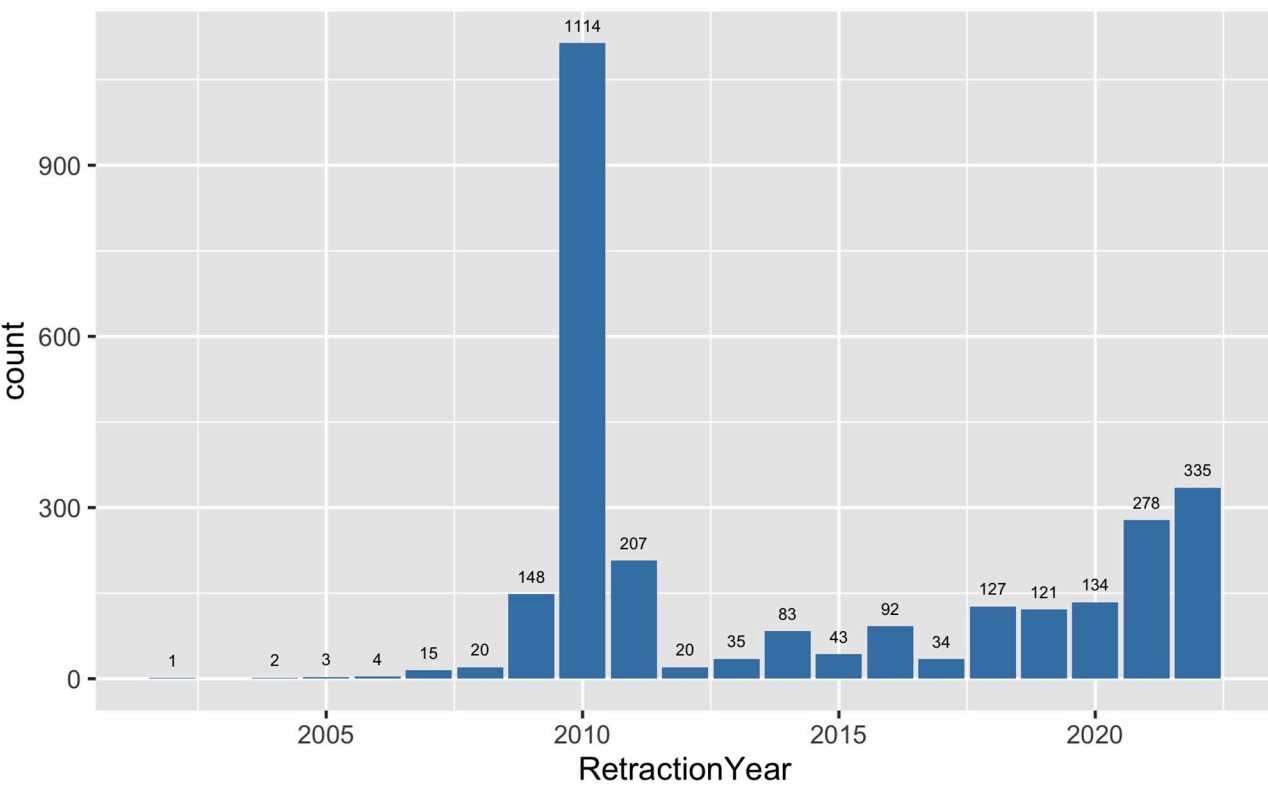

**Fig 1. Retraction year trends.** Retraction counts by year in Computer Science (NB The count for 2022 is incomplete).

classifies the paper in terms of discipline(s) and article type(s). Note that our analysis covers retractions only, so whilst important, EoCs and corrigenda are excluded.

### RQ1: The prevalence and nature of retraction in Computer Science

First, we describe the set of retracted CS papers. The retraction dates range over 20 years commencing in 2002 and the most recent being from May 2022 coinciding with the date of our version of the RW data set. Fig 1 reveals the trends with a general increase over time but with a pronounced spike from 2009–2011. The explanation is down to the mass retractions from several IEEE conferences that took place during this time period. For instance, the 2011 International Conference on E-Business and E-Government alone resulted in retractions of more than 1200 abstracts [3], although these are not limited to CS. Note also the probability of another spike for 2022 given we presently have data for 4.5 months only.

The median gap between publication and retraction is surprisingly low at 68 days although the skewed nature of the gap is apparent from the fact that the mean is 417 days and the most extreme value is 5497 days (approximately 15 years). Clearly, it is desirable for this gap to be as short as possible.

The RW database contains information on the subject field of a retracted paper. This is multi-valued so many papers are recorded as being related to multiple disciplines such as Computer Science *and* Education. When we retrieve all CS papers we actually obtain all papers that include CS however the subject count ranges from one to seven (see Fig 2). As we can see, the modal count is two, and only a minority (574/2816 = 20%) of papers are characterised as pertaining to uniquely CS. Interestingly, the CS proportion of single topic papers of 20% is

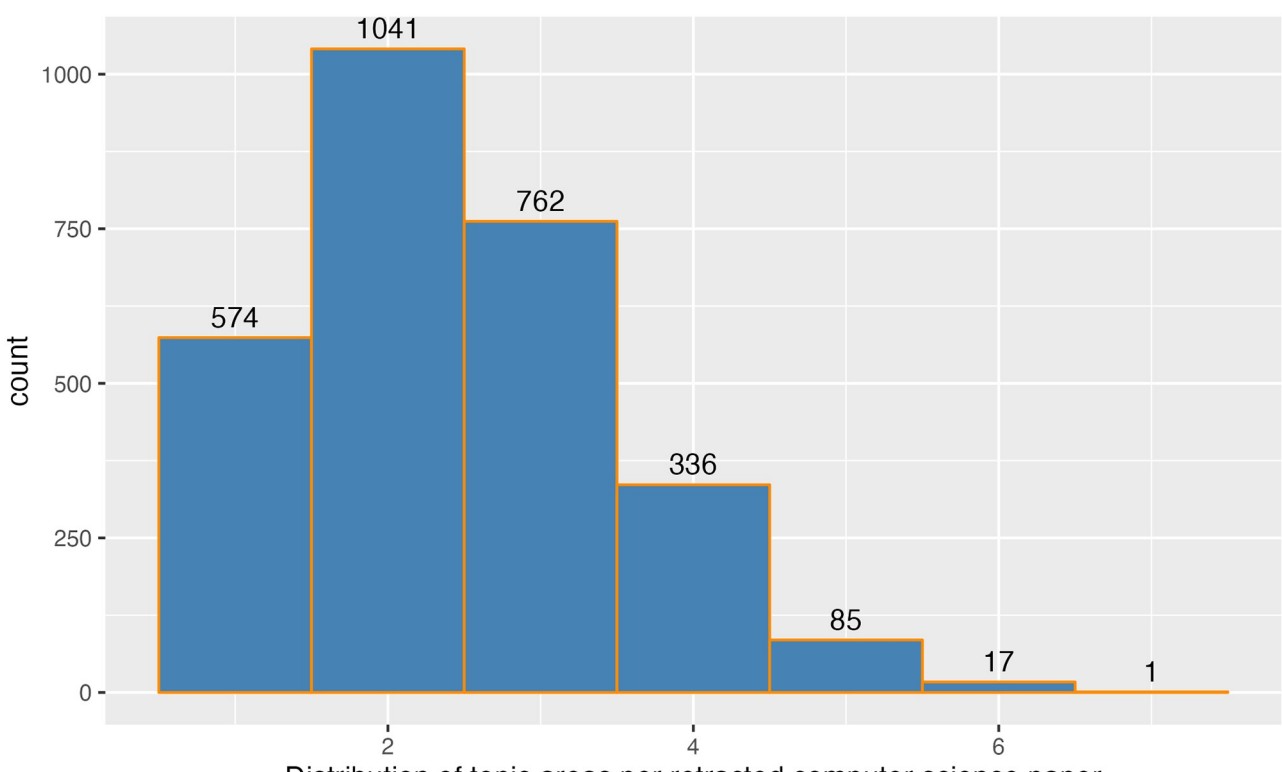

**Fig 2. Subject topic counts.** Histogram of subject topic counts for retracted Computer Science papers.

extremely close to the overall proportion of all retracted papers relating to a single topic i.e., 6618/33955 $\sim$ 19.5%. Informal inspection suggests these surprisingly high counts are due to (i) the application of CS to specific problem domains and (ii) a fine-grained set of subject topics. As an example, the retracted paper "An Improved Data Mining Technique for Classification and Detection of Breast Cancer from Mammograms" is classified under 7 topics:

```
(B/T)  Computer   Science;  (B/T)  Technology;
(BLS)  Biology  -  Cancer;(BLS)  Biology  -  Cellular;(BLS)  Neuroscience;
(HSC)  Medicine  -   Obstetrics/Gynecology; (HSC)  Medicine  -  Oncology;
```

Here we see that the seven subjects fall into three higher-level categories and indeed two categories since might be viewed as BLS and HSC are closely related. The only classification which is a little surprising is Neuroscience, however, this may be the consequence of the paper appearing in the journal *Neural Computing and Applications* where the 'neural' refers to a form of machine learning rather than neuroscience *per se*.

Next, we consider article type. Unfortunately, the database contains 80 distinct types of articles some of which seem to overlap and others are not relevant to Computer Science such as Clinical Studies. Specifically, there are 19 different article types for CS (the details are given in Table 1. However, there are some clear patterns. The largest category is Conference Abstract/Paper for which there are 1947 retractions, followed by 727 Research Articles. Some 19 papers are classified as Reviews or Meta-analyses (which is slightly fewer than the 28 we detected by searching in the title for either 'review' or 'meta-analysis'). The picture is somewhat

**Table 1. Frequencies of retracted Computer Science paper type.**

| Article type | Frequency |
|---|---|
| Conference Abstract/Paper | 1947 |
| Research Article | 727 |
| Conference Abstract/Paper;Research Article; | 28 |
| Article in Press;Research Article | 24 |
| Book Chapter/Reference Work | 21 |
| Review Article | 19 |
| Preprint | 15 |
| Article in Press | 10 |
| Others | 25 |

complicated by the fact that Article Type is another multi-valued attribute with some categories being composites such as "Article in Press; Research Article".

In terms of publishers, who in general manage the retraction process, again we see some quite pronounced differences. Overall there are a large number of different publishers many of which are institutional presses, societies, and so forth. We give the count of retractions for some of the better-known publishers in Table 2. We limit this list since the RW database lists 106 distinct publishers for CS alone. Note that article counts are taken from the Web of Science database and Total CS refers to the number of articles coded as Computer Science xxx or Information & Library Science. This may not exactly align with the codings from the RW database so the figures should be treated as approximate. What is most striking is the considerable disparity between publishers, with a tendency for the smaller publishers to have higher rates. Various factors may be relevant including reputation and also the phenomenon of bulk retraction typically of conference papers where the review process for the entire event is suspect. Nevertheless PLOS appears something of an outlier.

Next, we compare the distribution of retraction reasons in CS with all other disciplines (see Table 3). Note that the list of reasons is not exhaustive and there are many other reasons with low counts such as 'rogue' editors!

Fig 3 shows the relative proportions of retraction reasons for CS compared with other disciplines. Note that there can be multiple retraction reasons and also that we have coalesced closely

**Table 2. Comparison of retracted paper counts by selected Computer Science publishers.**

| Publisher | Total articles | Total CS articles | Retraction count | Retraction % |
|---|---|---|---|---|
| IEEE Explore | 3,800,000 | 1,477,000 | 1497 | 0.01 |
| IOP | 778,000 | 7,400 | 272 | 3.68 |
| SpringerLink | 6,577,000 | 726,000 | 236 | 0.03 |
| Elsevier | 13,780,000 | 443,000 | 160 | 0.04 |
| ACM Digital Library | 198,000 | 190,000 | 94 | 0.05 |
| Sage | 1,149,000 | 13,000 | 85 | 0.65 |
| Bentham | 84,000 | 681 | 50 | 7.34 |
| Hindawi | 255,000 | 18,900 | 32 | 0.17 |
| Taylor & Francis | 2,489,000 | 57,000 | 31 | 0.05 |
| Wiley | 6,233,000 | 72,000 | 17 | 0.02 |
| MDPI | 977,000 | 26,000 | 9 | 0.03 |
| PLOS | 316,000 | 792 | 6 | 0.76 |
| Overall | | ∼3,030,000 | ∼2,500 | ∼0.08 |

**Table 3. Comparison of the prevalence of retraction reasons between Computer Science and other disciplines.**

| Reason | Other | CS |
|---|---|---|
| Fake Review | 5.4% | 20.3% |
| Plagiarism | 12.7% | 8.0% |
| Duplication | 17.1% | 5.7% |
| Random paper | 1.6% | 14.3% |
| Problem author | 3.9% | 2.6% |
| Unresponsive author | 1.9% | 1.6% |
| Fake data | 4.6% | 0.1% |
| Problem data | 20.1% | 2.3% |
| Problem results | 13.3% | 2.9% |
| Problem analysis | 3.1% | 1.0% |
| Problem images | 2.7% | 0.0% |
| Inappropriate referencing | 2.0% | 12.1% |
| Withdrawn | 9.2% | 12.1% |
| Little or no information available | 26.0% | 55.9% |

related reasons into a smaller number of high-level categories. It is quite apparent that there are some non-trivial differences. We see four areas where CS exceeds other disciplines: randomly generated papers, fake reviews, referencing malpractice (such as citation cartels), and papers where the retraction request comes from some or all the authors. In contrast, for non-CS papers, problematic data, results, and duplicate and plagiarised papers dominated. Heibi and Peroni [25] have also noted how computer science seemed to be distinct from other disciplines.

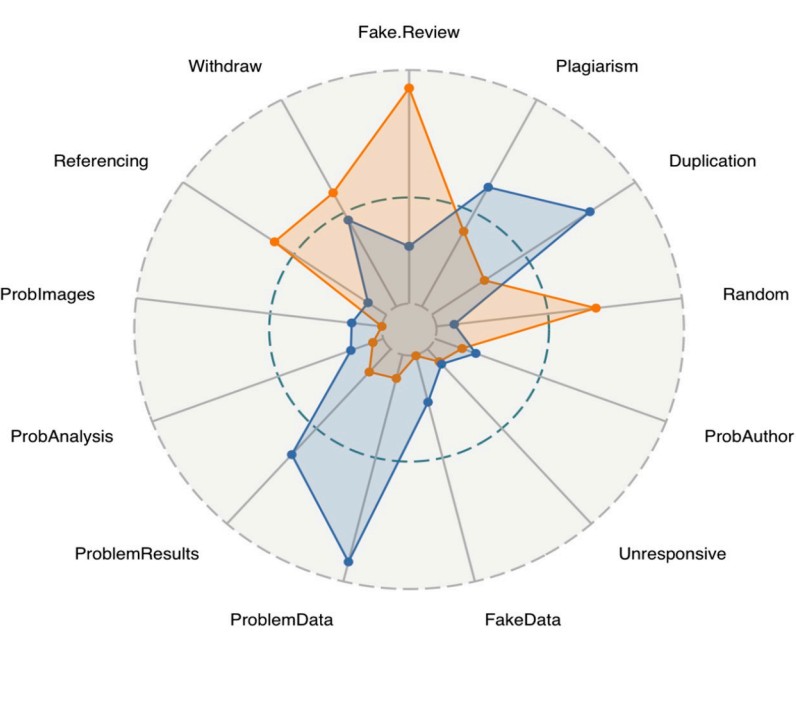

**Fig 3. Retraction reasons.** Radar plots show the relative proportions of retraction reasons by discipline.

However, the single, largest difference is not apparent from the chart in that for approximately 56% of CS papers, but only 26% of all other papers, little or no information is publicly available as to why the paper has been retracted. We do not show a lack of retraction reason on Fig 3 because it dominates the scale for the radar plot and compresses other retraction reasons to the point of unreadability. This seems problematic and also is a potential disservice to the authors since the reasons for retraction range from research misconduct to honest errors. In particular, the situation where the authors discover some problem with their paper and request its retraction (i.e., an honest error and a situation we would wish to encourage given the error has already been committed) will not be distinguished from more egregious situations of say data fabrication or plagiarism. Speculating, one possible reason is the prevalence of conferences in CS and the risk that the devolving of editorial and refereeing responsibilities to local groups could leave publishing vulnerable to exploitation and refereeing cartels. In such situations, publishers undertake bulk retractions and possibly make no distinction between papers leading to bland and uninformative retraction notices.

## RQ2: The post-retraction citation behaviour of retracted works

For our detailed analysis, we selected all retracted reviews including informal narrative reviews, systematic reviews, and meta-analyses. This amounted to 25 reviews with a further 4 papers being excluded due to not actually being a review of other work or, in one case, no longer available). We then undertook a stratified by retraction year random sample of a further 95 retracted non-review articles making a total of 120 articles. We resorted to sampling because an automated approach, for instance, using the R package Scholar, proved to be impractical due to our need to manually identify how many versions were available, language, determine and count meaningful citations, inconsistencies between publishers and so forth. Of these articles, 115/120 are available in the sense that the full text could be obtained using a Google Scholar search, albeit possibly behind a paywall. Ideally, all retracted articles should remain visible so that there is a public record of the history of the paper. The Committee on Publication Ethics (COPE) guidelines state that retracted papers should be labelled as retracted and be accessible (see https://publicationethics.org/files/cope-retraction-guidelines-v2.pdf. More worrying is the finding that *only* 98/120 ($\approx$ 82%) of the papers are clearly labelled as retracted in at least one, though not necessarily all, publicly available versions. It would seem that different publishers have different practices. Indeed individual journals and conferences may have different practices and these may evolve over time.

Another observation—and most likely relevant to the issue of post-retraction citations—is the proliferation of versions (see Fig 4). The median = 3 but as is clear from the histogram some papers have many more, with a maximum of 18. A feature of Google Scholar is it identifies links to multiple versions of a paper and often these will include informally published versions that might reside in the authors' institutional archive or repositories such as arXiv. The potential problem is that if there is no link to the VoR then even if the publisher marks a paper as retracted this decision may not, indeed frequently does not, propagate across all other versions thus the reader could be unaware of any retraction issues, or for that matter corrections.

Next, we identified all meaningful citations (i.e., from an English language, research article including pre-review items such as arXiv reports, also books, and dissertations) to the retracted paper. In total, our sample of 120 retracted papers was cited 1,472 times or put another way, potentially 'contaminated' 1,472 other CS papers. Since this represents less than 5% of all retracted papers (120/2,816) one can begin to imagine the impact. A very crude linear extrapolation based on the extrapolation is (2818/120) × 1472 ≈ 34543 might suggest that of the order of 30,000+ CS papers may be impacted by citations to retracted papers, although one possible

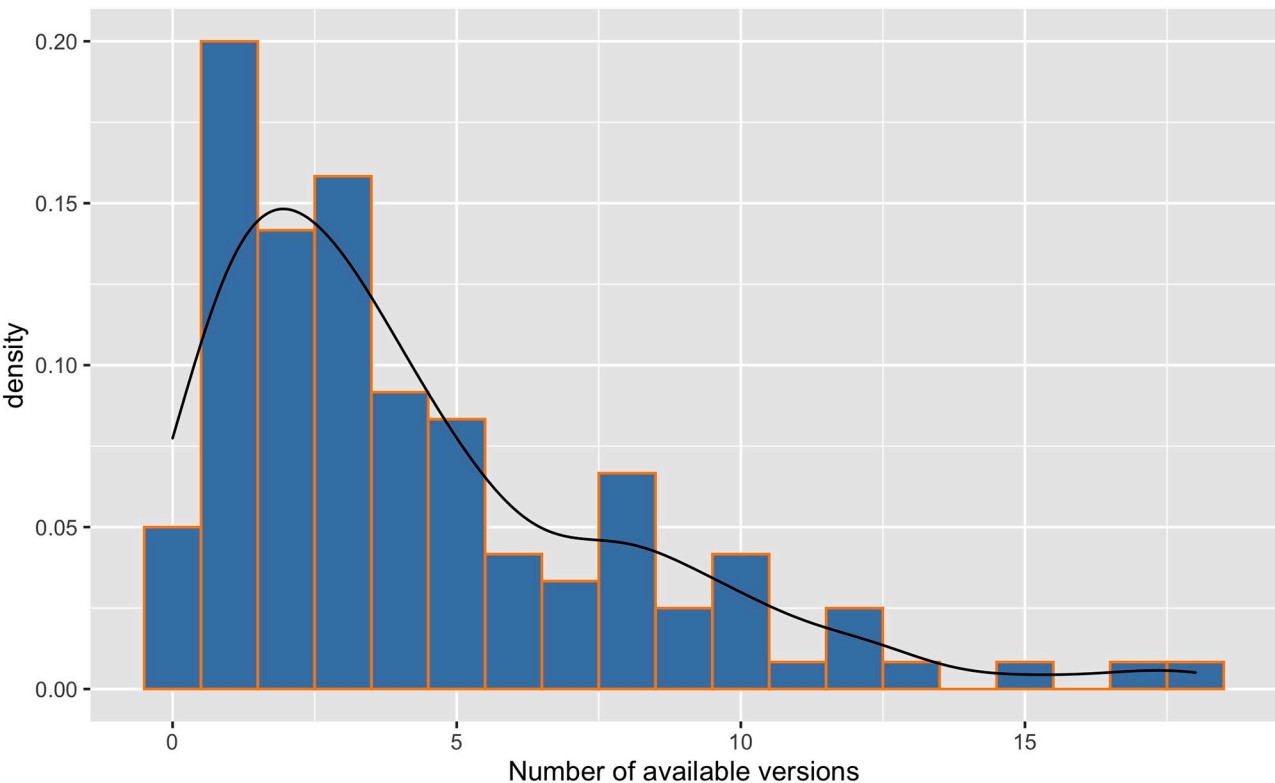

**Fig 4. Online versions.** Histogram and density plot of the available version count for 120 sampled CS retracted papers.

distortion is that our sample contains all the review articles which one might expect to be more heavily reviewed although this does not appear to be strongly the case (see the discussion for RQ3).

These citations to retracted CS papers were classified as being either before the year of retraction, during the year of retraction, or subsequent to the retraction i.e., post-retraction citations. Total citations ranged from zero to 145 with the median = 4, but the distribution is strongly positively skewed with the mean = 13.03. Unsurprisingly, we observed very disparate citation behaviours between the retracted CS papers, noting that of course, older papers have more opportunities to be cited. Interestingly the relationship between paper age and citation count is less strong than one might imagine, see Fig 6 plus there are other factors such as the venue and the topic.

Focusing now on post-retraction citations whilst allowing for concurrent (i.e., the citing paper is published at the same time as the cited paper is being retracted) submission, post-retraction is defined as being at least the following year to the retraction, we find a total of 672 post-retraction citations to 120 retracted papers (i.e., our sample). Again there is a huge disparity and these ranged from zero to a rather disturbing 82 with a median = 2 and a mean = 5.9 (see Fig 5). This is of course disturbing. It suggests either poor scholarship or ineffective promulgation of the retraction notice. Recall that for 18% of our sample of 120 CS papers there was no clear indication that the paper had been retracted. Of course, one wonders about the remaining 82%.

Next, we consider the relationship with the year of retraction which is depicted by the scatter plot in Fig 6 along with a log-linear regression line and 95% confidence interval. This

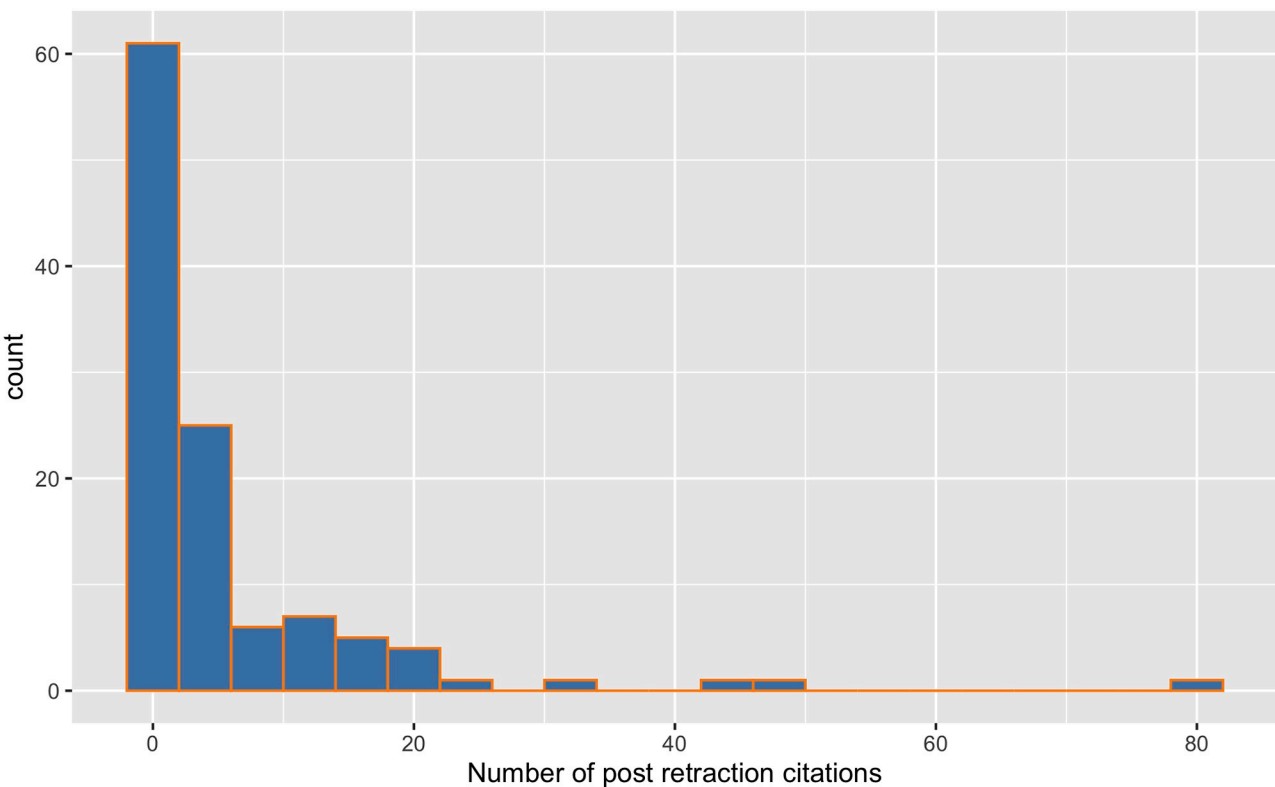

**Fig 5. Post-retraction citation counts.** Histogram of the distribution of the post-retraction citation count of 120 sampled papers.

suggests the relationship with age is surprisingly weak. We also indicate retracted review papers as distinct from regular papers but again the distinction is not strong (the respective medians when normalised by years since retraction are: review papers = 0.32 and non-reviews = 0.20).

Although not of much comfort, we observe that the post-retraction citation patterns for CS are not really distinct from other disciplines, and thus the problems of continuing to cite retracted papers are widespread and seem to run across all research communities [15, 17–19].

### RQ3: The potential impact upon systematic reviews and meta-analyses

Our final research question focuses on systematic reviews and meta-analyses because these articles are often highly influential. As the COPE Retraction Guidelines state "a]rticles that relied on subsequently retracted articles in reaching their own conclusions, such as systematic reviews or meta-analyses, may themselves need to be corrected or retracted".

First, we identified all retracted reviews (we located 28) via a search of the RW database using both their article classifications and an analysis of the titles. Then we analysed those papers that have cited a retracted review paper. Interestingly, there is no obvious difference in citation pattern with both types of paper having a median = 3 of post-retraction citations.

An alternative way of viewing this question is from the perspective of systematic reviews that cite retracted primary studies. Whilst the majority of our sample of retracted papers were not cited by any systematic review, we found that overall in our sample, approximately 30% of

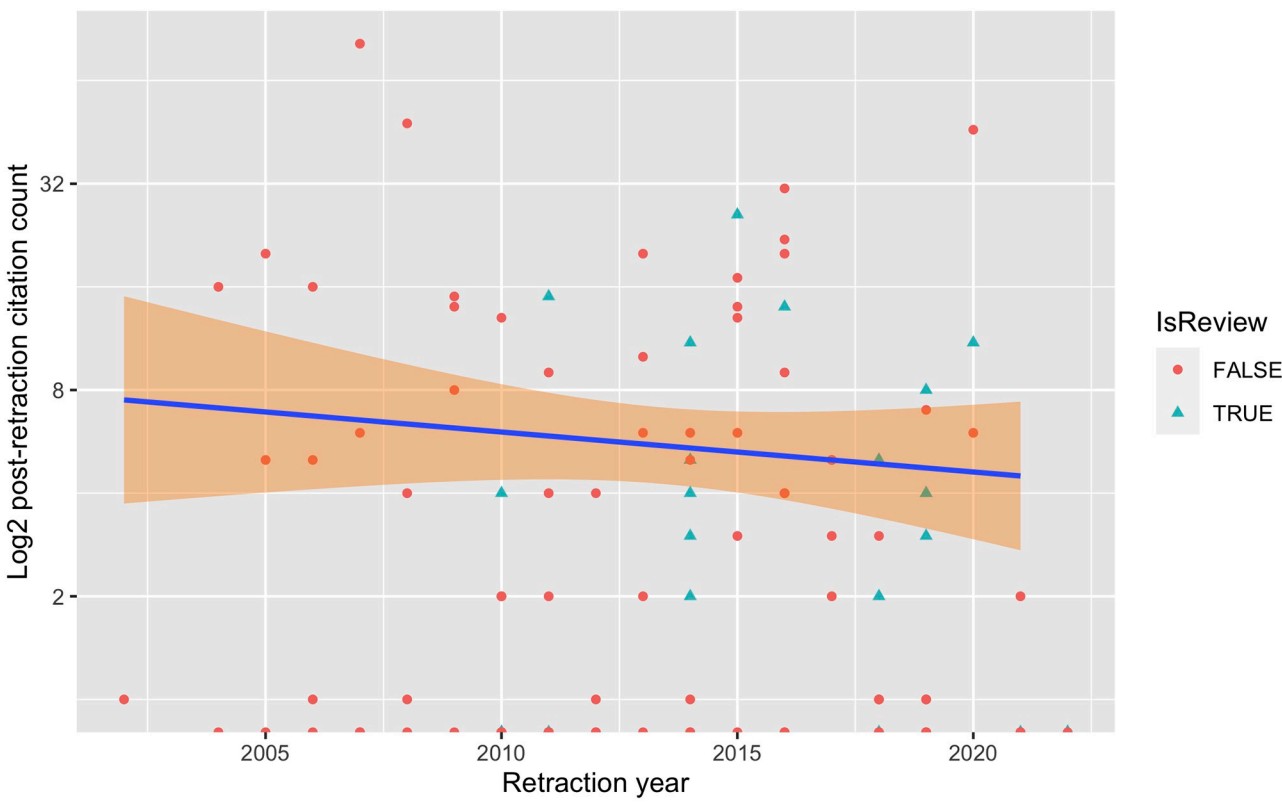

**Fig 6. Post-retraction citations by year.** Scatterplot of the distribution of the post-retraction citation by retraction year.

the retracted papers included one or more citations from a systematic review. The most extreme case was a highly-cited primary study that has a total of 93 citations including 31 since its retraction in 2016. Unfortunately, this study is included in four distinct systematic reviews and therefore its erroneous impact is quite far-reaching. It is beyond the scope of the present investigation to determine the impact of a retracted primary study on a systematic review though one can imagine it is likely to range from negligible to quite severe particularly given the tendency for lower-quality studies to report larger effect sizes [26].

In the course of analysing CS retractions and in particularly associated citations we did make a number of other observations.

1. As noted previously, there are markedly more retractions for the reason of randomly generated content than in other disciplines. In our sample of 120 papers, we noticed that the paper "Obstacle Avoidance Algorithms: A Review" has as one of its reasons for retraction is nonsensical content. Nevertheless, it has 7 citations none of which are self-citations so this rather suggests some authors are citing works without reading them. However, in general, we can be encouraged by finding that such papers are less cited (16 nonsense papers are only cited 9 times compared with the remaining 104 papers being cited 1465 times).

2. A couple of papers were retracted by the authors and then it seems corrected and replaced e.g., "Semantic Domains for Combining Probability and Non-Determinism". Here we replaced the citation analysis with NA because in all probability the citers are reasonably citing a new corrected version of the paper with the same title. The only point is that this process is hardly transparent.

3. A particularly flagrant example of attempting to 'game' the bibliometrics system, is the self-citation of the paper "Simple and efficient ANN model proposed for the temperature dependence of EDFA gain based on experimental results" retracted in 2013 from a 2021 paper. Perhaps this is an example of forgiving and forgetting?!

## Summary and recommendations

We summarise our findings as follows:

1. Most (76/120) of our sample of CS retracted papers continue to be cited at least one year after their retraction.

2. Although in the majority of cases ($\approx$82%) the official VoR is clearly labelled as retracted, the proliferation of copies e.g., on institutional archives, may not be so. Researchers relying on bibliographic tools such as Google Scholar are likely to be particularly vulnerable to being unaware of a paper's retraction status.

3. CS seems to lag behind other disciplines in terms of publicly providing the retraction reasons with approximately 56% of retracted papers offering little or no explanation.

4. Practice between different publishers can vary widely in terms of rates and provision of retraction reasons.

5. Producers of systematic reviews and meta-analyses need to be particularly vigilant. It has also been argued that citing retracted papers can be an indicator of poor scholarship and therefore a useful quality indicator when assessing candidate primary studies for inclusion. Of course "citing a retracted paper is not necessarily an error, but it is poor practice to cite such a paper without noting that it was retracted" [27].

   Specifically, we recommend that:

1. journals should follow a "standardised nomenclature that would give more details in retraction and correction notices" [3]

2. as researchers we become vigilant to the possibility that studies we wish to cite to support our research and arguments may have been retracted

3. community-wide consideration needs to be given for mechanisms to update papers, especially reviews and meta-analyses, that are impacted by retracted papers

4. the active adoption of tool support to help researchers better identify retracted papers e.g., the welcome integration of the RW database into the Zotero bibliographic management tool (see https://www.zotero.org/blog/retracted-item-notifications/. On this note, a recent study by Avenell et al. [28] suggests that proactive mechanisms are likely to be required. They investigated 88 citing papers of 28 retracted articles and despite between 1-3 emails notifying the authors, after 12 months only 9/88 $\approx$ 10% had published notifications.

### Threats to validity

**Internal threats** relate to the extent to which the design of our investigation enables us to claim there is evidence to support our findings. Measurement error is a possibility not least because as a simplifying assumption we treat all citations as equal, whilst [21] suggests that

there are multiple reasons for citation. Against this, Bar-Ilan and Halevi [15] found in their analysis of 238 citations to the 15 most cited but retracted papers (1995-2014) from the Elsevier ScienceDirect database that negative citations are rare and do not well predict retracted papers. Of course, a caveat is that 15 papers are a small sample. Ultimately we believe that whilst every retracted paper and its citations are in some sense its own story, we still strongly believe that whatever the context, high levels of citations to retracted papers are a considerable cause for concern.

The classification of papers/identification of reviews is imperfect e.g., we discovered one of our review papers from the random sample that was neither flagged as a review by RW. Note that the RetractionWatch database has been subsequently updated. Thus it is possible that other such papers have been missed.

**External threats** concerns the generalisability of our findings. Here, we argue that since the Retraction Watch database aims to be exhaustive overall our situation is more akin to a census than a sample which clearly reduces the difficulties of claiming representativeness. The remaining issue relates to our sample of citation data where we deployed a stratified procedure. Although we sampled by year, it is clearly possible that the relatively small sample (120 out of 2,816 observations) might be unrepresentative and one could gain confidence by increasing the sample size. This is an area for follow-up work.

## Further work

Whilst, we encourage other scholars and researchers to reproduce and extend our analysis and to that end uur R code is available as an R Markdown file from https://doi.org/10.5281/zenodo.6634462 however, we are unable to make the Retraction Watch Database available due to a data use agreement that prohibits publishing more than 2% of the data set. This requirement arises because, in order to fund Retraction Watch's continued operations, given that their initial grants have ended, they are licensing their data to commercial entities. Therefore researchers will need to approach Retraction Watch directly to obtain the same data set. Our understanding is that permission will not be withheld for *bona fide* research work.

A key theme and one which currently is under-explored is the role of multiple versions of retracted papers being available, and not all versions being under the aegis of the publisher. One driver for this situation might be the growing emphasis on Open Science and the expectation of researcher finders that versions of papers should not be behind paywalls. A widely used search engine that can locate any version is Google Scholar. The question is could such organisations cooperate in flagging all versions of a retracted paper as such?

Another theme that would benefit from the further investigation is the low proportion of meaningful retraction notices in CS compared to other disciplines. It is clear that meaningful reasons for retraction would be of value to the community and potentially to the authors especially when the reasons fall into the category of 'honest error'. Although we speculate that one driver could be the many conferences and subsequent bulk retractions it would clearly be valuable to delve deeper.

Complementary work would be to explore individual retractions in more depth, essentially as case studies. In a descriptive study, different subgroups of authors might be identified by applying clustering methods to the group of retracted papers during the time. In order to construct meaningful explanations of authors' subgroups based on the reason for retraction in a precise prediction, the integration of meta-analysis and supervised learning along with

Bayesian models might be utilised to uncover the impact of the retraction categories on the number of papers published by a particular author.

Another area to develop is to dig deeper into the specific retraction reasons since different retraction reasons have different causes and consequences. Also, there are more subcategories from the retracted reason categories than our analysis has explicitly covered since we have focused on the more commonly occurring categories.

## Acknowledgments

Martin Shepperd was supported by the Swedish Research Council as the recipient of the 2022 Tage Erlander research professorship and by a sabbatical from Brunel University London, although not specifically for this research. Both authors thank Retraction Watch for sharing their data set and the helpful comments on an earlier draft received from Ivan Oransky and Alison Abritis. They also thank the journal reviewers and editor for their helpful suggestions.

## Author Contributions

**Conceptualization:** Martin Shepperd, Leila Yousefi.

**Data curation:** Martin Shepperd, Leila Yousefi.

**Formal analysis:** Martin Shepperd, Leila Yousefi.

**Investigation:** Martin Shepperd.

**Software:** Martin Shepperd, Leila Yousefi.

**Writing – original draft:** Martin Shepperd, Leila Yousefi.

**Writing – review & editing:** Martin Shepperd, Leila Yousefi.

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
