## [Decision Letter · Decision Letter 0]

11 Sep 2022

PONE-D-22-17180An analysis of retracted papers in Computer SciencePLOS ONE

Dear Dr. Shepperd,

Thank you for submitting your manuscript to PLOS ONE. After careful consideration, we feel that it has merit but does not fully meet PLOS ONE’s publication criteria as it currently stands. Therefore, we invite you to submit a minor revision of the manuscript that addresses the points raised during the review process.

We look forward to receiving your revised manuscript.

Kind regards,

Jacopo Soldani

Academic Editor

PLOS ONE

Journal Requirements:

“Martin Shepperd was supported by the Swedish Research Council as the recipient of the 2022 Tage Erlander research professorship and by a sabbatical from Brunel University London.”

Reviewers' comments:

Reviewer's Responses to Questions

**Comments to the Author**

1. Is the manuscript technically sound, and do the data support the conclusions?

Reviewer #1: Yes

Reviewer #2: Partly

2. Has the statistical analysis been performed appropriately and rigorously? 

Reviewer #1: Yes

Reviewer #2: Yes

3. Have the authors made all data underlying the findings in their manuscript fully available?

Reviewer #1: Yes

Reviewer #2: No

4. Is the manuscript presented in an intelligible fashion and written in standard English?

Reviewer #1: Yes

Reviewer #2: Yes

5. Review Comments to the Author

Reviewer #1: This work analyzis retracted papers in Computer Science (taken from the Retraction Watch database) and their citations (taken from Web of Science and Google scholar). The aim of the work is investigating 3 main research topics: (RQ1) the prevalence and nature of retraction in Computer Science, (RQ2) The post-retraction citation behavior of retracted works, (RQ3) The potential impact upon systematic reviews and meta-analyses.

I really enjoyed reading this paper. Indeed, it is well written and structured which makes it easy to follow/understand and enjoyable to read. The analyzed aspects and the presented findings are very interesting. Therefore, my final review regarding this work is extremely positive.

I have just one main suggestion: considering the fact that in some cases retraction watch classifies the retracted papers under more than just one subject. e.g., a retracted paper in CS might be classified under Medicine as well. I think discussing a little bit more this fact, or even present some stats regarding the subjects that appeared the most together with CS will enrich the overall findings and discussion.

Other comments/suggestions for each section are listed below:

Background

+ Please cite the document defining the reasons of retraction you have listed: “… “A non-exhaustive list includes: …“

+ I like the discussion regarding the reason of citation (or citation function), you might consider including other relevant articles, such as: (1) “Teufel, S., Siddharthan, A., & Tidhar, D. (2006). Automatic classification of citation function. Proceedings of the 2006 Conference on Empirical Methods in Natural Language Processing - EMNLP ’06, 103. https://doi.org/10.3115/1610075.1610091”, (2) “Heibi, I., & Peroni, S. (2022). A protocol to gather, characterize and analyze incoming citations of retracted articles. PLOS ONE, 17(7), e0270872. https://doi.org/10.1371/journal.pone.0270872”  to classify the citation reason. (3) “Tuarob, S., Kang, S. W., Wettayakorn, P., Pornprasit, C., Sachati, T., Hassan, S.-U., & Haddawy, P. (2020). Automatic Classification of Algorithm Citation Functions in Scientific Literature. IEEE Transactions on Knowledge and Data Engineering, 32(10), 1881–1896. https://doi.org/10.1109/TKDE.2019.2913376”

Analysis and Results

+ while discussing the “article type”, I think you should talk a little bit more (and refer to relevant previous studies) about what is the most frequent publishing typology in Computer Science. This will let us have the right perception regarding the numbers of retraction for each different typology in CS.

+ You have mentioned that CS has a high number of retractions (65.3%) reporting “Little or no information available” compared to the general trend (27.4%). This fact itself could be considered for a whole new study, yet, can you discuss it more (based on your experience and impressions) in the conclusions (or as part of a “Further work”) .

+ The analysis toward “the proliferation of versions” is very interesting, Well done!

+ “… we noticed that the paper ”Obstacle Avoidance Algorithms: A Review” has …” Although you talk about a retracted paper, citing it is not prohibited, as long as it is clearly stated the fact that the article is retracted in both the text and its reference entry (as it has been suggested also by Retraction Watch, see https://retractionwatch.com/2018/01/05/ask-retraction-watch-ok-cite-retracted-paper/#:~:text=It's%20perfectly%20fine%20to%20cite,retracted%20papers%20in%20our%20database.). This is still of course a personal decision, yet, I think citing a retracted article is completely fine, the more delicate aspect is regarding “how it is cited“ (of course in a world where citations treated only from a quantitative point of view are used to rank authors and journals, this fact is very delicate and indeed a subject of future discussion). You have other similar situations in the rest of your paper.

Summary and Recommendations

+ “Although in the majority of cases (≈ 82%) the official VoR is clearly labelled as retracted, the proliferation of copies e.g., on institutional archives, may not be so. Researchers relying on bibliographic tools such as Google Scholar are likely to be particularly vulnerable to being unaware of a paper’s retraction status.”, This is indeed a very interesting aspect!

+ ““However, Bar-Ilan and Halevi [13] found that negative citations are rare and do not well predict retracted papers. … whatever the context, high levels of citations to retracted papers are a considerable cause for concern.” the work you are citing here analyzed a specific use case, so be careful in generalizing their findings to other domains and case study, such as the one you are analyzing (CS). Your following statement is very strong, I suggest you to reword this sentence in a “supposition“ style.

Reviewer #2: 1. On page 2, the US Office for Research Integrity’s definition for research misconduct was stated but no intext citation was given and the appropriate document from US Office for Research Integrity was not included in the list of references.

2. Instead of subsuming the methodology used in addressing each research question under the section on Analysis and Results, the authors should add a section on methodology, where methods used in extracting data from Retraction Watch as well as the methods used in analyzing the collected data are described.

3. On page 13, recommendation #4 needs a little bit of explanation.

4. The manuscript lacks clarity in a few areas and contains a few typographical and grammatical errors. It would therefore benefit from some editing.

6. PLOS authors have the option to publish the peer review history of their article (what does this mean?). If published, this will include your full peer review and any attached files.

Reviewer #1: No

Reviewer #2: No

---

## [Author Response · Author response to Decision Letter 0]

23 Oct 2022

We have provided a response to each comment in the response to reviewer (and editor) comments document entitled PLOS-One-ResponseToReviewers.pdf

---

## [Decision Letter · Decision Letter 1]

16 Jan 2023

PONE-D-22-17180R1An analysis of retracted papers in Computer SciencePLOS ONE

Dear Dr. Shepperd,

Thank you for submitting your manuscript to PLOS ONE. After careful consideration, we feel that it has merit but does not fully meet PLOS ONE’s publication criteria as it currently stands. Therefore, we invite you to submit a revised version of the manuscript that addresses the points raised during the review process. Please see a few minor comments from our in-house Staff Editors under "Additional Editor Comments".

We look forward to receiving your revised manuscript.

Kind regards,

Hanna Landenmark

Staff Editor, PLOS ONE

on behalf of

Jacopo Soldani

Journal Requirements:

Additional Editor Comments (if provided):

Notes from Staff Editor Hanna Landenmark (hlandenmark@plos.org):

1) Please note that PLOS ONE does not publish footnotes. We thus ask that you either move these comments into the main text, or delete any footnotes that you do not want to be published.

2) Line 370: "Practice between different publishers varies widely" - we do not feel that this conclusion is supported by the results, and that this phrasing would need amending.

3) Table 2 does not include information on publication volume, which we feel could be made clearer using additional information, such as reporting of a percentage or similar.

Reviewers' comments:

Reviewer's Responses to Questions

**Comments to the Author**

1. If the authors have adequately addressed your comments raised in a previous round of review and you feel that this manuscript is now acceptable for publication, you may indicate that here to bypass the “Comments to the Author” section, enter your conflict of interest statement in the “Confidential to Editor” section, and submit your "Accept" recommendation.

Reviewer #1: All comments have been addressed

Reviewer #2: All comments have been addressed

2. Is the manuscript technically sound, and do the data support the conclusions?

Reviewer #1: (No Response)

Reviewer #2: Yes

3. Has the statistical analysis been performed appropriately and rigorously? 

Reviewer #1: Yes

Reviewer #2: Yes

4. Have the authors made all data underlying the findings in their manuscript fully available?

Reviewer #1: Yes

Reviewer #2: No

5. Is the manuscript presented in an intelligible fashion and written in standard English?

Reviewer #1: Yes

Reviewer #2: Yes

6. Review Comments to the Author

Reviewer #1: (No Response)

Reviewer #2: (No Response)

7. PLOS authors have the option to publish the peer review history of their article (what does this mean?). If published, this will include your full peer review and any attached files.

Reviewer #1: No

Reviewer #2: No

---

## [Author Response · Author response to Decision Letter 1]

22 Feb 2023

Notes from Staff Editor Hanna Landenmark (hlandenmark@plos.org):

1) Please note that PLOS ONE does not publish footnotes. We thus ask that you either move these comments into the main text, or delete any footnotes that you do not want to be published.

CHANGED: All footnotes incorporated into text.

2) Line 370: "Practice between different publishers varies widely" - we do not feel that this conclusion is supported by the results, and that this phrasing would need amending.

CHANGED: "Practice between different publishers can vary widely in terms of rates and provision of retraction reasons."

CHANGED: Abstract: changed 'remarkable' to [t]here is also 'some' disparity.

3) Table 2 does not include information on publication volume, which we feel could be made clearer using additional information, such as reporting of a percentage or similar.

CHANGED: The table has been extended with additional columns on estimated total count of articles and estimated percentage of retracted articles. The data are extracted from the Web of Science database. We also add the following commentary.

"This may not exactly align with the codings from the RW database so the figures should be treated as approximate. What is most striking is the considerable disparity between publishers, with a tendency for the smaller publishers to have higher rates. Various factors may be relevant including reputation and also the phenomenon of bulk retraction typically of conference papers where the review process for the entire event is suspect. Nevertheless PLOS appears something of an outlier."

---

## [Editor Report · Decision Letter 2]

24 Apr 2023

An analysis of retracted papers in Computer Science

PONE-D-22-17180R2

Dear Dr. Shepperd,

We’re pleased to inform you that your manuscript has been judged scientifically suitable for publication and will be formally accepted for publication once it meets all outstanding technical requirements.

Kind regards,

Jacopo Soldani

Academic Editor

PLOS ONE
---

## [Editor Report · Acceptance letter]

28 Apr 2023

PONE-D-22-17180R2 

An analysis of retracted papers in Computer Science 

Dear Dr. Shepperd:

I'm pleased to inform you that your manuscript has been deemed suitable for publication in PLOS ONE. Congratulations! Your manuscript is now with our production department. 

Kind regards, 

on behalf of

Dr. Jacopo Soldani 

Academic Editor

PLOS ONE